# Improving the robustness of the Sequentially Optimized Reconstruction Strategy (SORS) for visual field testing

**Runjie Bill Shi**[1,2]*, **Moshe Eizenman**[3], **Yan Li**[4], **Willy Wong**[1,4]

**1** Institute of Biomedical Engineering, University of Toronto, Toronto, Canada, **2** Temerty Faculty of Medicine, University of Toronto, Toronto, Canada, **3** Department of Ophthalmology and Vision Sciences, University of Toronto, Toronto, Canada, **4** The Edward S. Rogers Sr. Department of Electrical & Computer Engineering, University of Toronto, Toronto, Canada

* bill.shi@mail.utoronto.ca

**Data Availability Statement:** The dataset used in this study are from the third party Rotterdam Ophthalmic Data Repository. Such visual field data is accessible online via the following link: http://

## Abstract

Perimetry, or visual field test, estimates differential light sensitivity thresholds across many locations in the visual field (e.g., 54 locations in the 24–2 grid). Recent developments have shown that an entire visual field may be relatively accurately reconstructed from measurements of a subset of these locations using a linear regression model. Here, we show that incorporating a dimensionality reduction layer can improve the robustness of this reconstruction. Specifically, we propose to use principal component analysis to transform the training dataset to a lower dimensional representation and then use this representation to reconstruct the visual field. We named our new reconstruction method the transformed-target principal component regression (TTPCR). When trained on a large dataset, our new method yielded results comparable with the original linear regression method, demonstrating that there is no underfitting associated with parameter reduction. However, when trained on a small dataset, our new method used on average 22% fewer trials to reach the same error. Our results suggest that dimensionality reduction techniques can improve the robustness of visual field testing reconstruction algorithms.

## Introduction

Standard automated perimetry, or visual field test, is an important psychophysical test that assess the functional integrity of the retina and visual pathway. It reports the differential light sensitivity thresholds across the retina in units of decibels (dB) at standard grid locations. Perimetry is most commonly used for the diagnosis and monitoring of glaucoma, as well as for other neuro-ophthalmological conditions that may cause visual field defects.

The 24–2 pattern is a commonly used visual field test pattern that consists of a grid of 54 locations. Typically, 3 to 7 stimulus presentations are required at each individual location to determine its threshold. Thus, a full 24–2 visual field test takes approximately 200 to 350 trials (3 to 6 minutes) per eye. In today's automated perimeters, these trials are decided by a computer algorithm, or "strategy." For example, one strategy is the staircase method in which

www.rodrep.com/longitudinal-glaucomatous-vf-data—description.html. The authors do not have ownership of the data and the data is available to all interested researchers under the same terms by which it was made available to the authors.

**Funding:** RS received the Canada Graduate Scholarships – Doctoral (CGS D) award by the Natural Sciences and Engineering Research Council of Canada (NSERC). Details of the award can be found at: https://www.nserc-crsng.gc.ca/students-etudiants/pg-cs/cgsd-bescd_eng.asp The funders had no role in study design, data collection and analysis, decision to publish, or preparation of the manuscript.

**Competing interests:** The authors have declared that no competing interests exist.

subsequent stimuli are made brighter/dimmer in constant step sizes depending on the subject's responses. The intensity at which a reversal in response occurs is reported as the estimated threshold.

Estimating thresholds at 54 locations takes a significant amount of time using traditional methods. Because a shorter test duration is desirable, the central design goal of perimetry algorithms has been to provide accurate and precise estimates using as few trials as possible. One approach to save time is quadrant-based "seeding" [1,2]. Quadrant seeding exploits the fact that thresholds within the same quadrant often have similar thresholds (after adjusting for the normal hill of vision). In the initial phase of the test, the thresholds at the four center locations of each quadrant are determined first. Their results are then used to initialize the staircase procedures at other locations in the same quadrant. Compared to no seeding, quadrant seeding reduces the overall test duration by 15% to 25% for a mixed glaucomatous/healthy dataset.

Kucur and Sznitman recently introduced a new data-driven seeding algorithm (referred to as a "meta-strategy"), called Sequentially Optimized Reconstruction Strategy (SORS) [3–5]. SORS reconstructs the visual field using a linear regression model by training on a large dataset. In other words, this method provides a model for reconstructing an entire visual field from just a subset of the field. Compared to quadrant-based seeding, SORS has much greater flexibility and is optimized by fitting to a real visual field training dataset. For example, when using SORS, the initial first four seeding locations are not constrained to be at the center of each quadrant. Instead, SORS is data-driven and the testing sequence (which location is tested first, second, third, etc.) is learned automatically. It optimizes the reconstruction model to provide the best reconstruction error without manual specification or explicit domain knowledge. Moreover, seeding/reconstruction extends beyond the first four locations. Estimates of the whole field are incrementally improved whenever new locations are tested. Given the accurate reconstruction provided by SORS, the authors proposed to terminate the test after testing 36 locations, with comparable accuracy to traditional algorithms, and the test duration is cut by about 30% as a result [3].

The original SORS uses a sequence of ordinary linear regression models for reconstruction. This process requires specifying a large number of parameters due to the high dimensional representation of visual field data (i.e., 54-dimensional for a 24–2 visual field). A common issue with large machine-learning models with lots of parameters is overfitting. Overfitting occurs when a trained model is too closely aligned to a limited set of training data points. When provided with a large training dataset, SORS has been shown to perform well. However, with smaller training datasets overfitting may become a problem if the model is not robust.

Clinical practice suggests that visual field data may be much lower dimensional in nature. Clinicians recognize common patterns of glaucomatous visual field defects, such as nasal step, arcuate defect, tunnel vision, etc. Furthermore, glaucomatous field loss often predominantly involves either the superior or the inferior hemi-field [6]. Visual field test locations have been divided previously into a few sectors based on their anatomical relationships with regions of the optic nerve head. Such sector-based approaches have provided more robust analyses of visual field thresholds [7,8]. Therefore, a visual field may be seen as a superposition of these common patterns, rather than 54 independent variables.

If visual field data is indeed lower dimensional, a compressed, lower-dimensional description of visual fields may be automatically derived through mathematical techniques such as principal component analysis (PCA). Dimensionality reduction techniques have been used previously as feature extraction techniques in visual field post-processing or classification tasks [9–14]. We argue that they may also be used to address overfitting in high-dimensional reconstruction models because a lower-dimensional embedding reduces the overall number of

model parameters. To the best of our knowledge, dimensionality reduction has not been applied directly to the process of visual field testing.

In this paper, we investigate two different dimensionality reduction algorithms as an additional step in the reconstruction model: principal component analysis (PCA) and partial least squares (PLS). We incorporate these methods and compare the performance against existing methods (e.g., the original SORS linear regression method) by carrying out simulations across both large and small training sets and with subjects of varying reliability levels. Our simulation results demonstrate that dimensionality reduction methods can significantly improve the robustness of the reconstruction model in a small dataset.

## Methods

PCA and PLS for visual field reconstruction is compared against the existing ordinary linear regression. The methodologies as well as the details concerning the simulations are described in this section.

### Sequentially Optimized Reconstruction Strategy (SORS)

Without loss of generality, we assume that there are 54 test locations (24–2 grid) in a visual field, denoted as $x \in \mathbb{R}^{54}$. To train SORS, we assume a series of pre-determined thresholds and select the next location to test by minimizing the training reconstruction error of the whole field. This error is calculated using the tested locations and the additionally selected location to reconstruct the visual field and computing the point-wise mean squared error. The process is repeated until all locations have been exhausted. For full details, please refer to the original paper by Kucur and Sznitman [3].

Note that this training method is a greedy algorithm that does not guarantee the exact global optimal solution (which would require an infeasible, exhaustive search of all possible combinations and permutations). Nonetheless, Kucur and Sznitman showed that this algorithm outperforms other viable alternatives [3].

At test time, visual field simulations can be performed by testing one location at a time. However, in real-world applications, testing must be done in small batches (e.g., every four locations [4]) to avoid querying the same location consecutively. Our simulations here simulated this latter approach, but the results are not significantly different from testing one location at a time.

### Reconstruction Models for SORS

Three reconstruction models are now described. These methods include the ordinary linear regression method (used in the original SORS), as well as two dimensionality reduction techniques to be compared with ordinary linear regression.

**Linear Regression (LR).**  The original SORS paper used an ordinary linear regression model for the reconstruction of visual fields [3]. The reconstructed visual field estimate is denoted as $\hat{\mathbf{x}} \in \mathbb{R}^{54}$ such that:

$$\hat{\boldsymbol{x}} = D\boldsymbol{s} + \boldsymbol{\beta} \qquad\qquad \text{Eq 1}$$

Here $\boldsymbol{s}$ represents a vector of thresholds that have already been determined thus far into the test (a subset of the whole visual field). $D \in \mathbb{R}^{54 \times S}$ and $\boldsymbol{\beta} \in \mathbb{R}^{54}$ represent the weights and bias trained by linear regression mapping $\boldsymbol{s}$ to a prediction of reconstructed visual field $\hat{\boldsymbol{x}}$.

**Transformed-Target Principal Component Regression (TTCPR).**  The linear regression model above does not use any feature extraction or processing. In the transformed-target

principal component regression (TTPCR) algorithm, we add principal component analysis to the process to reduce the number of weights and dimensionality of the model. Suppose there is a training dataset matrix $X \in \mathbb{R}^{54 \times N}$ of $N$ visual fields each with 54 thresholds. The training is broken into two distinct steps.

The first step in the training process is to reduce the training matrix $X$ from $54 \times N$ to $n \times N$, where $n \ll 54$, using principal component analysis (PCA). We denote this reduced low-dimensional matrix as $T \in \mathbb{R}^{54 \times n}$ and refer to it as the "embedding" of the visual field data and $n$ is the "embedding dimension." The exact PCA algorithm to convert $X$ to $T$ is described in the next few paragraphs.

The training matrix $X$ consists of column vectors of visual field examples, i.e., $X = [\boldsymbol{x}_1, \boldsymbol{x}_2, \ldots, \boldsymbol{x}_N]$. Let $B \in \mathbb{R}^{54 \times N}$ be the original matrix with mean, $\boldsymbol{\mu}$, subtracted from each column (i.e., each visual field) in $X$.

$$B = [\, \boldsymbol{x}_1 - \boldsymbol{\mu}, \; \boldsymbol{x}_2 - \boldsymbol{\mu}, \; \ldots, \; \boldsymbol{x}_N - \boldsymbol{\mu} \,] \qquad \text{Eq 2}$$

Note that $B$ has the same dimensions as $X$. Let the covariance matrix $S \in \mathbb{R}^{54 \times 54}$ be defined as:

$$S = \frac{1}{N-1} B B^T \qquad \text{Eq 3}$$

$S$ is always a symmetric matrix. There exists an eigendecomposition such that

$$S = \frac{1}{N-1} V \Lambda V^T \qquad \text{Eq 4}$$

where $\Lambda \in \mathbb{R}^{54 \times 54}$ is a diagonal matrix of non-increasing eigenvalues of $S$ and corresponding columns of $V \in \mathbb{R}^{54 \times 54}$ are eigenvectors of $S$ of non-increasing significance. The top four eigenvectors (i.e., first four columns of $V$) in the Rotterdam glaucomatous visual field dataset are visualized later in results and may be interpreted as representing archetypical visual field patterns in the dataset ("eigen-visual fields"). The significance of these fields will be discussed later.

Let $W$ be the matrix constructed by taking the top $n$ eigenvectors (first $n$ column vectors) of $V$, i.e., $W = [\boldsymbol{v}_1, \boldsymbol{v}_2, \ldots, \boldsymbol{v}_n] \in \mathbb{R}^{54 \times n}$. Finally, $T$ can be calculated from the original training dataset as:

$$T = W^T B \in \mathbb{R}^{n \times N} \qquad \text{Eq 5}$$

The second step in the training process is to train a linear regression model between the input $\boldsymbol{s}$ and output $\hat{\boldsymbol{t}}$, where $\hat{\boldsymbol{t}}$ is an estimate of embedding (column vector in $T$) reconstructed from a vector of thresholds that have already been determined thus far into the test $\boldsymbol{s}$.

$$\hat{\boldsymbol{t}} = D\boldsymbol{s} + \boldsymbol{\beta} \qquad \text{Eq 6}$$

This is similar to Eq 1 above. The difference is that in Eq 1, the linear regression model estimates the visual fields $\hat{\boldsymbol{x}}$ directly, while in the TTPCR the linear regression model only estimates the embeddings $\hat{\boldsymbol{t}}$. These embeddings are then mapped to a full visual field estimate $\hat{\boldsymbol{x}}$ using the PCA relationships that were learnt during training. That is, at test time, the visual field is reconstructed as:

$$\hat{\boldsymbol{x}} = W\hat{\boldsymbol{t}} + \boldsymbol{\mu} = W(D\boldsymbol{s} + \boldsymbol{\beta}) + \boldsymbol{\mu} \qquad \text{Eq 7}$$

We specifically named this model "transformed-target" principal component regression to avoid confusion with standard principal component regression where the input is transformed by PCA. In our case, the target output is transformed by PCA and the linear regression models are fitted to reconstruct the embeddings.

The TTPCR structure includes embeddings extraction and linear regression. The TTPCR structure is fast to train because it has a closed-form solution, but the embeddings extracted are not guaranteed to be optimal for the reconstruction task.

**Partial Least Squares (PLS) regression.** Partial least squares (PLS) is a different dimensionality reduction technique which computes the embeddings by directly minimizing the reconstruction error. Therefore, it can provide theoretically better performance than TTPCR. PLS uses the following model:

$$\hat{x} = WDs + \beta \qquad\qquad \text{Eq 8}$$

Where $D \in \mathbb{R}^{n \times S}$ and $W \in \mathbb{R}^{54 \times n}$ are two matrices with constrained rank $n \ll 54$ to be trained. Here $n$ is the embedding dimension and is equivalent to the $n$ used in PCA above.

Unlike TTPCR, PLS does not have closed-form optimal solution for $W$ or $D$, so it requires iterative training and is slower. The PLS structure resembles a two-layer artificial neural network without non-linear activation.

The network architectures of LR, TTPCR, and PLS are illustrated in Fig 1.

**Comparison baseline models.** We implemented two additional methods as baseline models for comparison.

First, we simulated a situation where no seeding/reconstruction procedure was used, and each location was considered independently. To be consistent with the SORS framework, the values at untested locations were simply set by the expected values of the hill of vision. These

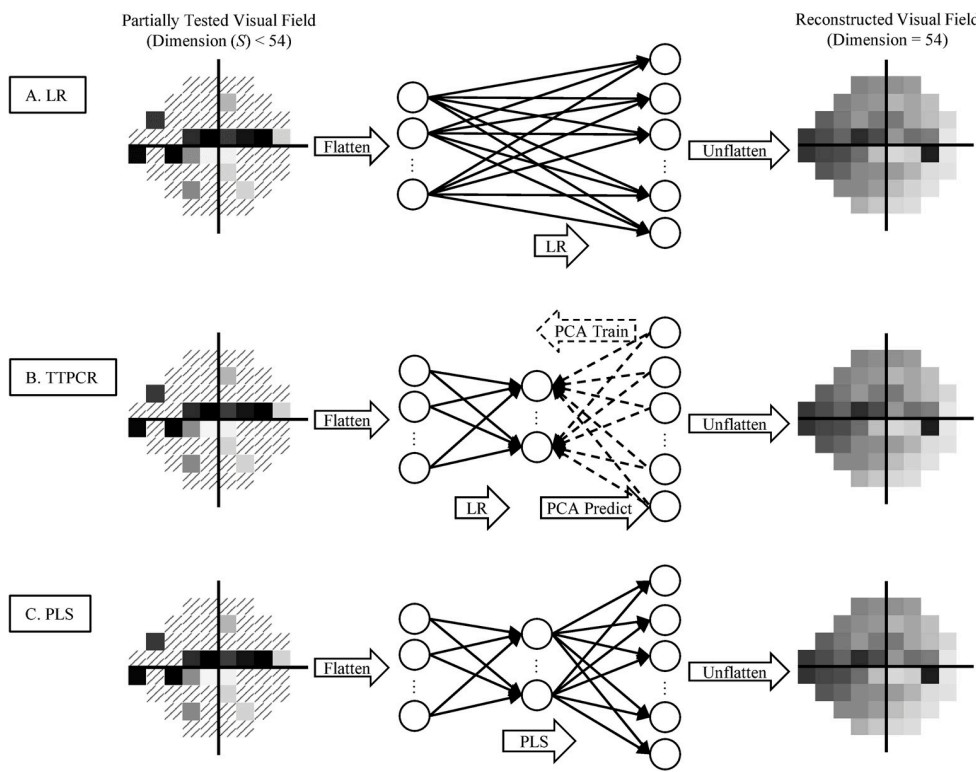

**Fig 1. Illustration of the linear regression (LR), transformed-target principal component regression (TTPCR), and partial least squares (PLS) methods.** Note that while only one general diagram is shown for each model, for SORS in fact many models are trained until the optimal model is found for reconstruction from S = 1, 2, . . ., 53 to 54 locations. In TTPCR, the second PCA layer is trained once and frozen for all cases of S. This contrasts with PLS, where both layers are optimized for each model.

values were derived as the mean of the sensitivity thresholds at these locations in the training dataset. We denoted this as the "mean" (hill of vision) reconstruction method.

Second, we simulated the commonly used "quadrant" seeding method. In this method, the centers of the four quadrants (±9˚, ±9˚) were tested first. Then, the tested locations' neighbors started with the same deviation from the expected normal hill of vision as the already tested quadrants' locations. This was described in more detail by Turpin et al. [2] (See Turpin's Fig 1).

## Simulation

We trained and examined the reconstruction error using SORS together with LR, TTPCR, PLS, mean, and quadrant reconstruction models using the 278 eyes with 24–2 visual fields in the Rotterdam longitudinal glaucomatous visual field dataset, which is the same dataset as in previous work [3,15]. The Rotterdam dataset encompasses a wide variety of glaucomatous patterns and has repeated measurements, ensuring the representativeness of the dataset to real-life conditions.

The point-wise root-mean-square error (RMSE) of a visual field estimate is defined as:

$$\text{RMSE} = \sqrt{\frac{1}{54} \sum_{i=1}^{54} (x_i - \hat{x}_i)^2} \qquad \text{Eq 9}$$

Where $x_i$ and $\hat{x}_i$ are the true and estimated/reconstructed thresholds at location $i$, respectively. Point-wise RMSE is used to evaluate the accuracy of the result, while number of trials is used for test duration.

Two cross-validation approaches were trained and evaluated:

1. Small training set with 5-fold cross-validation (using 20% for training and 80% for testing);

2. Large training set with 10-fold cross-validation (using 90% for training and 10% for testing).

These two conditions were used to investigate any signs of overfitting or underfitting by the testing methods. Overfitting can occur due to the large number of parameters used in the SORS linear regression model, making the model fit too closely to the training data but in case of overfitting it will fail to generalize to unseen testing data. Overfitting may be observed in the small training dataset when the algorithm sees only a limited number of visual fields. Underfitting is the reverse problem where there are too few parameters to fit the actual data and can be observed when there are large errors in both the training and testing datasets. Finally, there is a difference in the number of folds between the two training sets. While the number of folds affects the variance of the results, it will be clear from our results that this does not pose a problem.

Threshold estimation at a single location was implemented using a Bayesian method (Zippy Estimation by Sequential Testing, ZEST) [2,16,17]. (Additional results using 4–2 Staircase are shown in the S1 and S2 Figs) The implementations are consistent with previous studies [2]. The prior probability mass function was empirically derived from histograms of the input training data [2,16,17]. The ZEST strategy was set to terminate after the standard deviation reaches <2.0 dB and the mean of the probability mass function was returned as the estimate.

We present Monte Carlo visual field testing simulation results from a reliable responder with 3% false positive rate and 3% false negative rate. (Additionally, results from less reliable responders with 15% false positive rate and 3% false negative rate, as well as 15% false positive rate and 15% false negative rate, are included in the S3 and S4 Figs) The psychometric function was a piece-wise linear function approximation as previously described by Turpin et al. [2].

Linear regression, PCA, Transformed Target Regressor, PLS, and the "mean" reconstruction method ("Dummy Regressor") were implemented using Scikit-learn version 1.0.2 [18]. Implementations of these visual field strategies can be found in our open-source PyVF library available at https://vf.shirunjie.com. The dataset to reproduce the results is publicly available in the Rotterdam Ophthalmic Data Repository at http://www.rodrep.com/data-sets.html.

## Results

This section details Monte Carlo simulation results with cross-validation using the Rotterdam dataset. First, we explore how the results vary with embedding dimension. After establishing an optimal embedding dimension, we examine the error of reconstruction as a function of the number of tested locations. Finally, we compare the sequence of testing locations generated from different reconstruction methods.

### Effect of embedding dimension on SORS-TTPCR and SORS-PLS testing error

Fig 2 shows the performance after testing 36 locations using the SORS strategy with TTPCR and PLS versus the embedding dimension (the embedding dimension refers to either the number of principal components in PCA or the dimension of the intermediate representation in PLS) [19].

The left panel shows the case when a small training dataset is used (20% of the full dataset or about 56 eyes). The optimal testing error is achieved with an embedding dimension of around 8–12. With larger embedding, even though the training error remains the same, the testing error actually increases with additional parameters, indicating overfitting [19]. In the limit when the embedding dimension is equal to 54 (i.e., same as the dimensionality of the 24–2 field), both TTPCR and PLS become mathematically equivalent to LR, clearly demonstrating that overfitting occurs for SORS with LR when trained with a small training dataset.

The right panel shows when a large training dataset is used (90% of the full dataset or about 250 eyes). In this case, there is no systematic overfitting even when 54 dimensions are used. The original SORS paper only evaluated this case of a large training dataset.

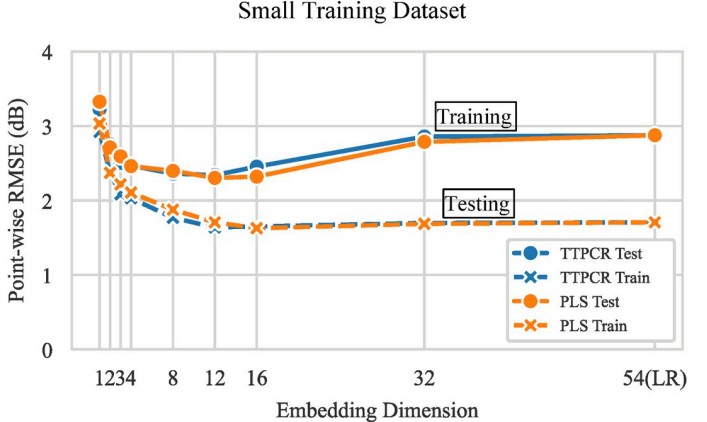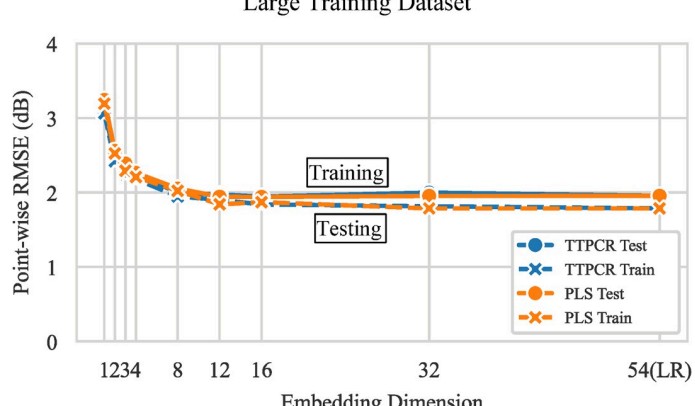

**Fig 2. Point-wise RMSE as a function of the embedding dimension (number of principal components) with TTPCR and PLS after testing 36 locations in a reliable responder (3% false positive and 3% false negative).** (Left) Cross-validation results with a small (20%) training dataset. Testing error is much larger than training error when the embedding dimension is large. (Right) Cross-validation results with a large (90%) training dataset. Testing error is similar to training error. TTPCR performs similarly to PLS in all cases.

From this, we can draw the following conclusions: (1) Both TTPCR and PLS yield approximately the same performance, despite the theoretical advantage of PLS. Since TTPCR is much faster to train (TTPCR has a closed-form solution while PLS must be trained iteratively), it is the better choice. (2) An embedding dimension of 8 with TTPCR is practically sufficient to achieve the best performance in both the small and large training dataset cases. (3) LR, which is equivalent to using the full 54 dimensions for TTPCR, performs well only with large training datasets but not with small training datasets. Since dimensionality reduction does not impede performance with large training datasets, it is suitable for general use and will work even when only small datasets are available.

## Comparison of SORS models

Next, we compare the performance of the different reconstruction models as a function of the number of determined thresholds. A model that achieves a lower reconstruction error with fewer determined locations is more efficient. Given the findings from the previous section, we will only present TTPCR with 8 dimensions in the following evaluations.

Fig 3 compares the testing performance of TTPCR and LR against the two baseline models ("Quadrant" and "Mean"). The simulation results show that the performance of TTPCR is robust against changes in the size of the training dataset. When using the large training dataset, after testing 36 locations, TTPCR achieves 2.00 dB point-wise RMSE in 139 trials, and this performance is essentially identical to that of LR (1.96 dB in 139 trials). (see gray dotted lines in Fig 3C and 3D) With the small training dataset, after testing 36 locations, TTPCR achieves 2.36 dB point-wise RMSE in 148 trials (see Fig 3A and 3B), which is not much worse than the performance when trained with the large dataset. As such, the dimensionality reduction technique used in TTPCR is robust to training dataset sizes and these results demonstrate that it can be used with a small training dataset with as few as 50–60 eyes.

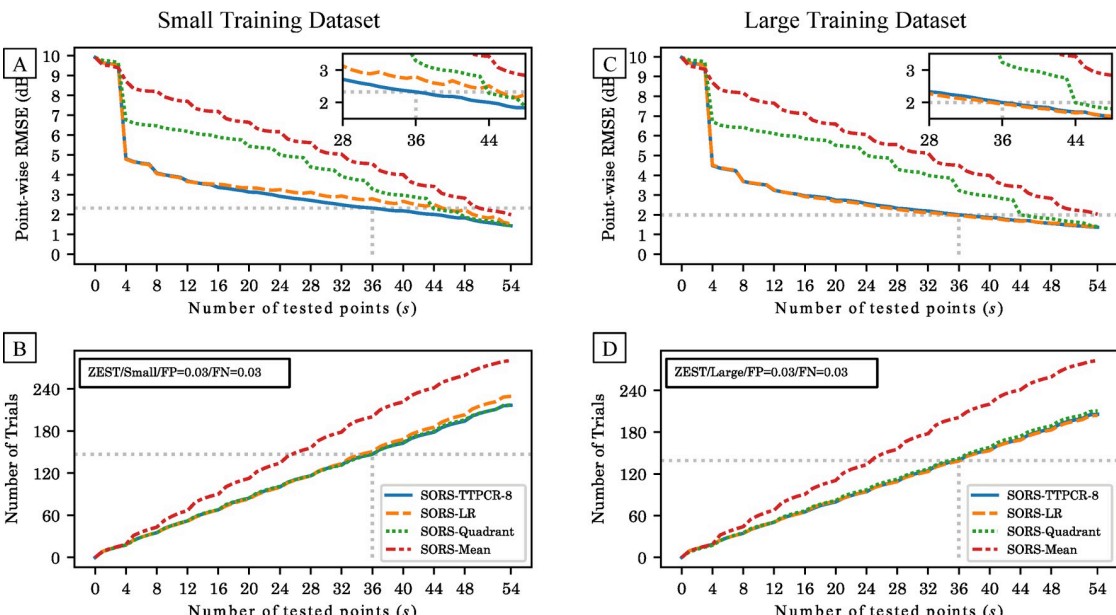

**Fig 3.** Point-wise RMSE (A and C) and number of trials (B and D) as a function of number of tested locations with TTPCR, LR, Quadrant, and Mean. Cross-validation results of using 20% for training (A and B, small dataset) and 90% for training (C and D, large dataset).

By contrast, the performance of LR is worse when using the small training dataset compared to using the large training dataset. After testing 36 locations, LR has a higher RMSE of 2.88 dB in more trials (152 trials). (See Fig 3A and 3B) In order to achieve the same 2.36 dB reconstruction error as TTPCR, LR would require testing 46 locations with 190 trials, making the test 29% longer. Similar observations can be drawn across all disease severity levels. Fig 4 shows the error and number of trials using a small training dataset for TTPCR(36), LR(36), and LR(46) (Note: the numbers 36 and 46 indicate the number of locations tested and used for reconstruction) in three severity groups: mild: MD>−6 dB; moderate: −12<MD≤−6 dB; severe: MD≤−12 dB. The same trend is observed across all three severity levels. This is explained by the fact that the additional parameters in LR reconstruction compared to TTPCR reconstruction require more training data.

Additionally, we have carried out the same simulation and analyses using the staircase thresholding strategy and arrive at the same conclusions. Please see the S1 and S2 Figs.

## Optimized test sequences

Unlike traditional testing methods where the test sequence is randomly shuffled at test time, the SORS method pre-determines the optimal sequence of locations for testing based on the training data. From this, we analyzed the optimal sequence for each method which is shown in Fig 5. Examining these different sequence, we make the following observations.

First, the quadrant-seeding growth map is significantly different from LR's test sequence. Quadrant-seeding tests the center of the quadrants (±9˚, ±9˚) first and then tests locations that neighbor these centers. This whole procedure is manually designed. With LR, the first four locations are sampled from four distinct quadrants, and the next four locations (5–8) are also from four distinct quadrants. However, these are not necessarily the centers of the quadrants

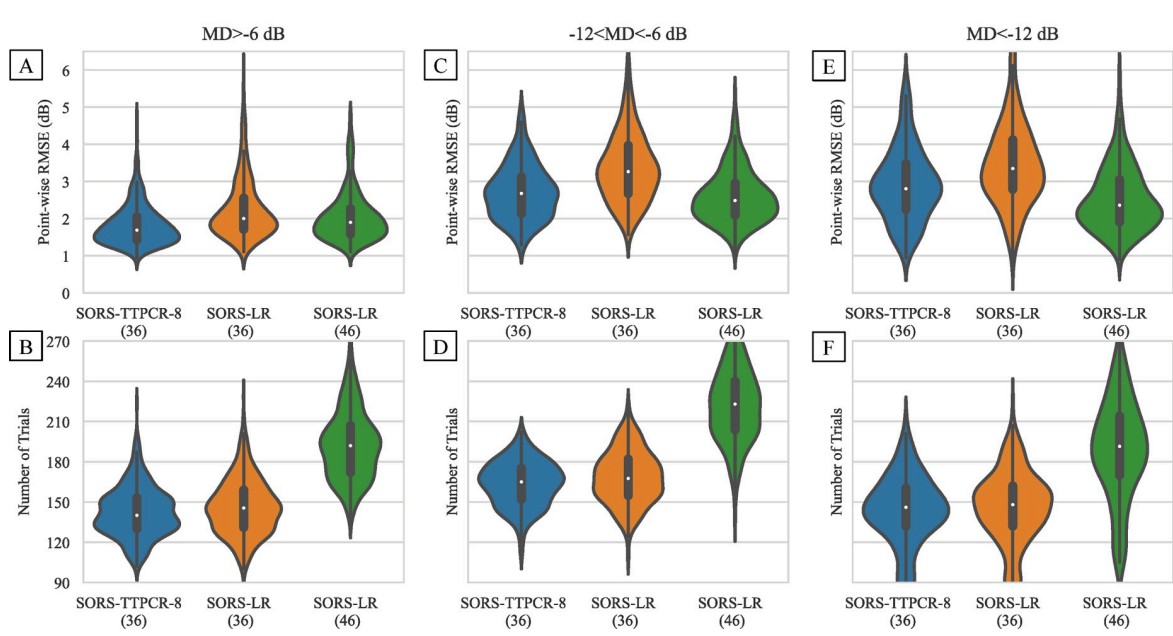

**Fig 4.** Distribution of point-wise RMSE (A, C, E) and test duration (B, D, F) after training on the small dataset using TTPCR and LR in different disease severities. When both are terminated after 36 locations, TTPCR(36) (blue) achieves lower RMSE than LR(36) (orange) (p<0.0001 for all severities). LR(46) (green) is able to achieve similar accuracy as TTPCR(36) (blue) but is 29% longer on average (p<0.0001 for all severities).

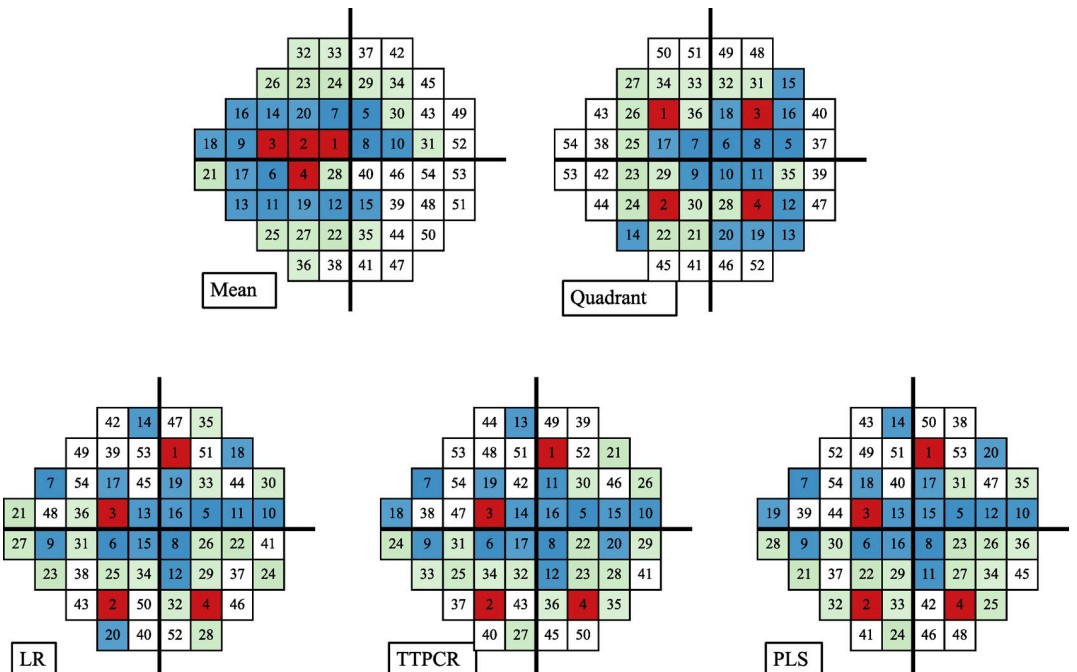

**Fig 5. Order of locations tested in the SORS algorithm using four different reconstruction models.** The first four locations are marked in red, the next 16 locations in blue, and the following 16 locations in green. Top row, left: Using the mean (hill-of-vision) estimator for untested locations in SORS essentially results in testing locations with the largest variances to locations with the smallest variances. Right: The traditional quadrant growth map first tests the four quadrant centers (marked in red), and then "grows" their deviations from normal values to neighboring locations. Bottom row, left to right: LR, TTPCR, and PLS models learn a sequentially optimized order of testing based on the training data. In the presented sequences, both TTPCR and PLS used an 8-dimensional embedding, while LR is 54-dimensional.

but are instead optimized empirically using the training data. As such, this suggests that there is some validity to the idea of quadrant seeding, but it is not likely optimal in terms of reconstruction error.

Second, despite differences in reconstruction models (LR, TTPCR, and PLS), the optimal test sequences across all methods are mostly identical. The first 20 locations (red and blue locations in Fig 5's LR, TTPCR and PLS) are almost the same. This similarity of test sequences demonstrates that, despite dimensionality reduction from 54 to 8 dimensions, TTPCR and PLS are able to reproduce the results from LR using much fewer parameters.

## Compressibility of visual fields and interpretation of principal components

Apart from improving the robustness of reconstruction models, another benefit of using dimensionality reduction technique is that it allow easy visualization and interpretation of the embedding layer. In PCA, the dataset is projected into a lower dimensional subspace defined by the principal components. These principal components represent the most important patterns that explain the most variance in the dataset. The right panel of Fig 6 visualizes the top 4 principal components. The first component seems to represent the contrast between the overall height of the field and the most common blind spot location in the 24–2 grid, (+15°, −3°). The second component seems to represent a general scotoma/depression pattern in the superior or the inferior hemifield. Note that the coefficient for each principal component can be positive or negative, so the second component is able to mathematically represent decreased sensitivity in either hemifield. The third and fourth components appear to bear some

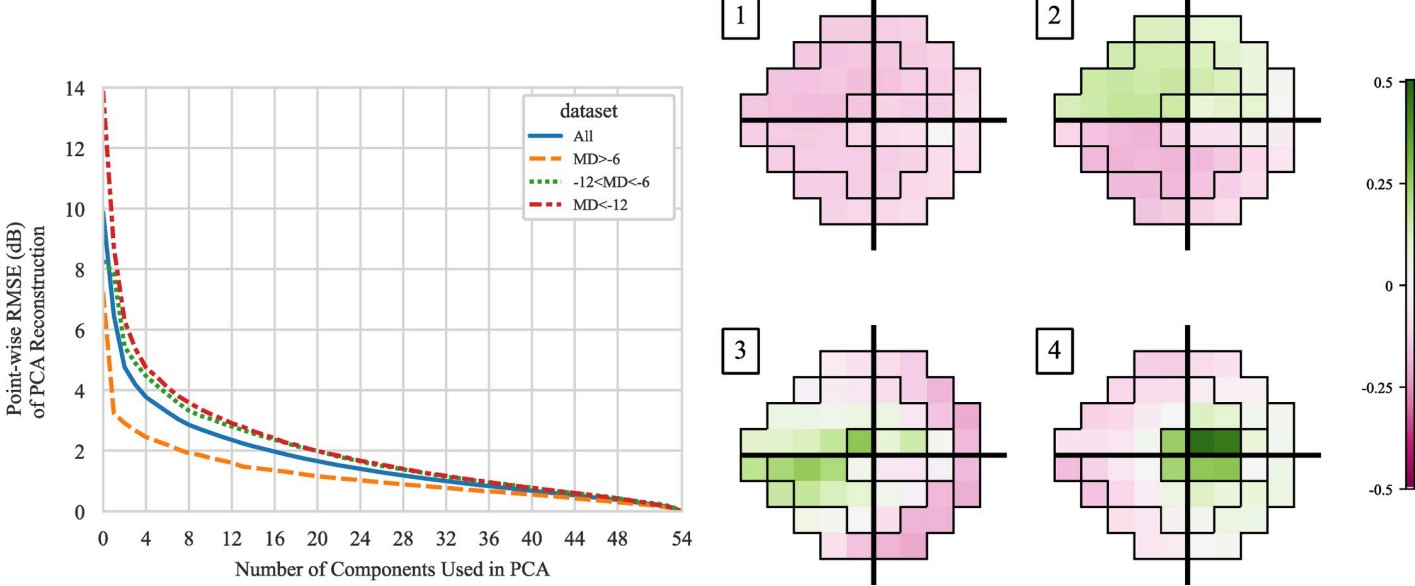

**Fig 6.** (Left) Point-wise RMSE of PCA reconstruction as a function of embedding dimension. Principal components were trained on the full dataset and then the PCA reconstruction error is evaluated on all, mild (MD>−6 dB), moderate (−12<MD<−6 dB), and severe (MD<−12 dB) data. (Right) Visualization of the top four principal components superimposed on the Garway-Heath visual field sectors.

resemblance to the Garway-Health visual field sectors [8]. These sectors were constructed based on their anatomical relationships with regions of the optic nerve head and have been used extensively in visual field post-analysis, but not utilized in visual field testing strategies. The PCA approach here is equivalent to the classical example of "eigen-faces" for facial recognition [20]. The principal components in Fig 6 may be thought of as "eigen-visual fields."

While using the first four components may capture most of the variance inside the visual field dataset, using more components will always capture more of the variance and achieve lower reconstruction error. However, the marginal benefit of adding more components diminishes. The left panel of Fig 6 shows this trend, i.e., the reconstruction error using 1, 2, 3, ..., 54 principal components to reconstruct the original dataset. A lower error suggests that the data can be represented using the principal component embedding with higher fidelity. Zero error can only be achieved by keeping all 54 components, but for a lossy compression, the marginal benefit of keeping the principal components past around 8–12 components is much lower than the first few components. This explains why an 8-dimensional embedding was found to be highly effective and efficient.

We also break down the reconstruction error by the severity of visual fields (different curves in the left panel of Fig 6). Unsurprisingly, healthier visual fields (orange) are easier to precisely reconstruct than visual fields with more significant defects (green and red). This is because mathematically a normal visual field that resembles the normal hill of vision only requires one parameter to describe, i.e., offset from a standard hill of vision. When using 8 principal components, the point-wise reconstruction RMSE when tested on all, mild, moderate, and severe visual fields are 2.9, 1.9, 3.3, and 3.6 dB, respectively.

## Discussion

In this paper, we introduced dimensionality reduction in the SORS meta-strategy's visual field reconstruction models. The original SORS approach used a 54-dimensional linear regression

model to reconstruct the 24–2 visual fields. We showed that this method can suffer from over-fitting when the training dataset is small. In our proposed TTPCR method, an 8-dimensional embedding was used to build a more robust reconstruction model that performs better and does not overfit for small training datasets, while performing similarly for large training data-sets. This low-dimensional embedding is obtained from the original visual field dataset using transformed-target PCA.

Visual field data, in its usual format, is high dimensional. In the case of a 24–2 testing grid, each visual field is a 54-dimensional vector. However, from clinical and anatomical observa-tions, we know that there are significant correlations between visual field locations. For exam-ple, in the Rotterdam longitudinal glaucomatous visual field dataset, thresholds of neighboring 24–2 locations in the same hemifield often have Pearson correlation coefficients that are greater than 0.9. Such high degree of linear correlations imply redundancy in glaucomatous visual field patterns, and thus a high degree of data compressibility. More rigorously, this redundancy was demonstrated in the left panel of Fig 6 where only a small number of principal components are necessary to achieve a good reconstruction of a visual field. The reconstruc-tion errors in mild, moderate, and severe eyes using 8 principal components are 1.9, 3.3, and 3.6 dB which are better than the expected visual field long-term variability (2.8, 4.3, and 4.0 dB) in the Rotterdam dataset.

Using this idea, we improved upon the ordinary LR model by investigating two dimension-ality-reducing methods that produce low dimensional embeddings in the reconstruction pro-cess, TTPCR and PLS. Both TTPCR and PLS, when selected with an appropriate number of dimensions (e.g., 8 dimensions) demonstrate robustness against overfitting regardless of patient reliability and disease severity and perform with the same efficacy as ordinary LR when there is a large dataset.

TTPCR was chosen over PLS due to the reduced training time while maintaining similar reconstruction performance. Unlike typical principal component regression where the input is transformed for the TTPCR we transformed the output (target) into principal component space. Since the training target remains the same during the entire SORS training, this princi-pal component transformation only needs to be computed once for the training target, so it does not impact the overall training time compared to LR. In comparison, PLS needs to be trained iteratively and is not guaranteed to converge quickly, making the training process much less efficient.

Both TTPCR and PLS are linear methods. We have also investigated non-linear dimension-ality-reducing methods, such as an artificial neural network-based autoencoder. Although such deeper, non-linear artificial neural network models are theoretically more powerful, it did not provide superior performance for the visual field reconstruction task. S5 Fig shows the reconstruction performance of a non-linear artificial neural network-based autoencoder. As can be seen, the performance is generally not better than the reconstruction performance of PCA (i.e., linear autoencoder) in the left panel of Fig 6. This suggests that even though non-lin-ear techniques can theoretically express much more complex relationships between the input and output, linear models are sufficient to capture most of the relationship between thresholds within a visual field. This also agrees with the aforementioned observation that the Pearson correlation between visual field thresholds at different locations are very high. A larger artifi-cial neural network, such as the ones used in more complicated algorithms [21], may provide a better performance, but defeats the practicality of a simple and robust method. Similarly, we have also tried ensembled-based methods, such as random forest and gradient boosting, which also has the capability to reduce overfitting, but they did not offer superior performance than TTPCR and PLS, and took much longer to train. While convolutional neural network may

seem like a natural choice for pricing local visual field patterns, the sparseness of the SORS sampling process prohibits any convolution operations.

Traditionally, improvements in visual field testing were made through developing more efficient single-location testing strategies. Bayesian strategies like ZEST are close to optimal so far as determining a single location's threshold is concerned. As a result, to obtain further reductions in testing times, many recent advances have come from exploiting the underlying spatial pattern of visual field thresholds. Quadrant seeding can be seen as the original idea in this direction by exploiting similarly in threshold deviation within the same quadrant. More recent ideas include spatially weighted likelihoods in ZEST (SWeLZ) [22], and spatial entropy pursuit (SEP) [23]. Both methods require the spatial relationships between locations to be manually pre-defined in a graph. In SWeLZ, the edge weight between two locations specifies the strength of relationship between these two locations. In SEP, the spatial graph is used to model the visual field as a "conditional random field" to compute estimates for untested locations.

SORS generalizes the idea of seeding to a data-driven paradigm. Unlike the strategies mentioned above, the spatial graph does not need to be manually specified, but, is automatically learned and optimized for reconstruction. Therefore, SORS can be rapidly deployed for reconstruction of other visual field patterns (i.e., for different diseases) and clinical usages. For example, for neuro-ophthalmology testing, one simply needs to re-train SORS on a corresponding dataset and does not need to manually re-design the strategy [24,25]. With the TTPCR reconstruction method proposed here, a small training dataset may be sufficient to achieve very good performance.

## Conclusions

The reconstruction-based visual field testing meta-strategy SORS can be further optimized through dimensionality reduction techniques like principle component analysis. In our proposed "transformed-target principal component regression" (TTPCR) model, the reconstruction of the visual field can be achieved robustly using only the top 8 principal components for a wide range of conditions without loss of accuracy when compared with the original ordinary linear regression model for SORS. Therefore, TTPCR can be used as a more efficient, robust replacement for ordinary linear regression in visual field reconstruction. Further work may wish to test this idea on visual field datasets of diseases other than glaucoma.

## Supporting information

**S1 Fig. Cross-validation performance using 4–2 staircase in a subject with FP = 3% and FN = 3%.**
(PDF)

**S2 Fig. Distribution of point-wise RMSE and test duration after training on a small dataset and using in different disease severities.**
(PDF)

**S3 Fig. Cross-validation performance using ZEST in a subject with FP = 15% and FN = 3%.**
(PDF)

**S4 Fig. Cross-validation performance using ZEST in a subject with FP = 15% and FN = 15%.**
(PDF)

**S5 Fig. Precision of autoencoder reconstruction on the Rotterdam dataset as a function of number of principal components used.** The autoencoder was trained on the full dataset and tested on all, mild (MD>−6 dB), moderate (−12<MD<−6 dB), and severe (MD<−12 dB) data. The autoencoder has the following architecture: input→54 hidden units→n-dimensional embedding→54 hidden units→reconstructed input.
(PDF)

## Acknowledgments

We would like to thank the Rotterdam Ophthalmic Data Repository for generously providing data sets related to ophthalmology for research.

## Author Contributions

**Conceptualization:** Runjie Bill Shi, Moshe Eizenman, Willy Wong.

**Data curation:** Runjie Bill Shi, Yan Li.

**Formal analysis:** Runjie Bill Shi.

**Funding acquisition:** Runjie Bill Shi, Willy Wong.

**Investigation:** Runjie Bill Shi, Yan Li, Willy Wong.

**Methodology:** Runjie Bill Shi, Moshe Eizenman, Yan Li, Willy Wong.

**Project administration:** Runjie Bill Shi, Moshe Eizenman, Willy Wong.

**Resources:** Runjie Bill Shi, Moshe Eizenman, Willy Wong.

**Software:** Runjie Bill Shi, Yan Li.

**Supervision:** Moshe Eizenman, Willy Wong.

**Validation:** Runjie Bill Shi, Yan Li, Willy Wong.

**Visualization:** Runjie Bill Shi.

**Writing – original draft:** Runjie Bill Shi, Moshe Eizenman, Yan Li, Willy Wong.

**Writing – review & editing:** Runjie Bill Shi, Moshe Eizenman, Yan Li, Willy Wong.

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
