## [Decision Letter · Decision Letter 0]

11 Sep 2023

PONE-D-23-14760Improving the Robustness of The Sequentially Optimized Reconstruction Strategy (SORS) for Visual Field TestingPLOS ONE

Dear Dr. Shi,

Thank you for submitting your manuscript to PLOS ONE. After careful consideration, we feel that it has merit but does not fully meet PLOS ONE’s publication criteria as it currently stands. Therefore, we invite you to submit a revised version of the manuscript that addresses the points raised during the review process.

We look forward to receiving your revised manuscript.

Kind regards,

Nouman Ali

Academic Editor

PLOS ONE

Journal Requirements:

2. Please note that PLOS ONE has specific guidelines on code sharing for submissions in which author-generated code underpins the findings in the manuscript. In these cases, all author-generated code must be made available without restrictions upon publication of the work. 

Please review our guidelines at https://journals.plos.org/plosone/s/materials-and-software-sharing#loc-sharing-code and ensure that your code is shared in a way that follows best practice and facilitates reproducibility and reuse.

"We would like to thank the support from Natural Sciences and Engineering Research Council of Canada (NSERC) and the Glaucoma Research Society of Canada.."

Funding information should not appear in the Acknowledgments section or other areas of your manuscript. We will only publish funding information present in the Funding Statement section of the online submission form. 

"RS received the Canada Graduate Scholarships – Doctoral (CGS D) award by the Natural Sciences and Engineering Research Council of Canada (NSERC). Details of the award can be found at: https://www.nserc-crsng.gc.ca/students-etudiants/pg-cs/cgsd-bescd_eng.asp The funders had no role in study design, data collection and analysis, decision to publish, or preparation of the manuscript."

Reviewers' comments:

Reviewer's Responses to Questions

**Comments to the Author**

1. Is the manuscript technically sound, and do the data support the conclusions?

Reviewer #1: Yes

Reviewer #2: Partly

2. Has the statistical analysis been performed appropriately and rigorously? 

Reviewer #1: Yes

Reviewer #2: No

3. Have the authors made all data underlying the findings in their manuscript fully available?

Reviewer #1: Yes

Reviewer #2: Yes

4. Is the manuscript presented in an intelligible fashion and written in standard English?

Reviewer #1: Yes

Reviewer #2: Yes

5. Review Comments to the Author

Reviewer #1: In the manuscript entitled “Improving the robustness of the sequentially optimized reconstruction strategy (SORS) for visual field testing”, the authors proposed a PCA-MLR-combined visual field reconstruction method. The research is scientifically sound and the manuscript is well organized. I have two minor questions for the authors. The first one is the performance comparison of the proposed method with the neural network model. In the comparison, the network structure was not clearly described. The node number in each layer and active function are also of importance for the performance. Please specify it. The other question is that why didn’t the authors try more commonly used machine learning models such as random forest or XGBoost? These methods were widely used and have shown encouraging performances.

Reviewer #2: Perimetry, or visual field test, estimates differential light sensitivity thresholds across many locations in the visual field (e.g., 54 locations in the 24-2 grid). Recent developments have shown that an entire visual field may be relatively accurately reconstructed from measurements of a subset of these locations using a linear regression model. Here, authors have shown that incorporating a dimensionality reduction layer can improve the robustness of this reconstruction. Specifically, they proposed to use principal component analysis to transform the training dataset to a lower dimensional representation and then use this representation to reconstruct the visual field. they named the new reconstruction method the transformed-target principal component regression (TTPCR). When trained on a large dataset, the new method yielded results comparable with the original linear regression method, demonstrating that there is no underfitting associated with parameter reduction. However, when trained on a small dataset, our new method used on average 22% fewer trials to reach the same error. The results suggest that dimensionality reduction techniques can improve the robustness of visual field testing reconstruction algorithms. Authors must address my following queries

• Presented values in figures and graphs must be statically verified.

• How the research used for comparison is selected in this manuscript, this is not clear

• How authors have selected the dataset in this research, it is not clear to readers

• Authors are suggested to cite and discuss

‘ Zheng, M., Zhi, K., Zeng, J., Tian, C., & You, L. (2022). A hybrid CNN for image denoising. Journal of Artificial Intelligence and Technology, 2(3), 93-99.’

‘ Zhang, Q., et al.: A robust deformed convolutional neural network (CNN) for image denoising. CAAI Trans. Intell. Technol. 1– 12 (2022). https://doi.org/10.1049/cit2.12110’

• Comparison with the baseline deep learning models is required along with the comparison with baseline dimensionality reduction techniques

• How are research variables selected for this manuscript?

6. PLOS authors have the option to publish the peer review history of their article (what does this mean?). If published, this will include your full peer review and any attached files.

Reviewer #1: No

Reviewer #2: No

---

## [Author Response · Author response to Decision Letter 0]

7 Nov 2023

Dear Editor and Reviewers, 

Thank you for reviewing our work. Below is a point-by-point comment on the reviewer's comments and the corresponding revisions. *Please note that since Figures are not able to be included in this response entry text input, the reviewers may view the uploaded Response to Reviewers for all non-text content and a better overall reading experience.*

Reviewer 1

> Reviewer #1: In the manuscript entitled “Improving the robustness of the sequentially optimized reconstruction strategy (SORS) for visual field testing”, the authors proposed a PCA-MLR-combined visual field reconstruction method. The research is scientifically sound and the manuscript is well organized. I have two minor questions for the authors. The first one is the performance comparison of the proposed method with the neural network model. 

>

> In the comparison, the network structure was not clearly described. The node number in each layer and active function are also of importance for the performance. Please specify it. 

We would like to thank Reviewer #1 for the constructive feedback. 

We have added Figure 1 in the main text which illustrates the network structure of the three methods compared in this paper: linear regression (LR, baseline method), transformed-target principal component regression (TTPCR), and partial least squares (PLS). The image is replicated here below. (*Note, please see the Response to Reviewer document or the revised manuscript for the figure.*)

Insert on line 164: “The network architectures of LR, TTPCR, and PLS are illustrated in Fig 1.”

(*Note, please see the Response to Reviewer document or the revised manuscript for the figure.*)

Fig 1. Illustration of the linear regression (LR), transformed-target principal component regression (TTPCR), and partial least squares (PLS) methods. Note that while only one general diagram is shown for each model, for SORS in fact many models are trained until the optimal model is found for reconstruction from S = 1, 2, …, 53 to 54 locations. In TTPCR, the second PCA layer is trained once and frozen for all cases of S. This contrasts with PLS, where both layers are optimized for each model.

> The other question is that why didn’t the authors try more commonly used machine learning models such as random forest or XGBoost? These methods were widely used and have shown encouraging performances.

In response to the Reviewer #1’s comments, we have trained and tested two ensemble methods, random forest and gradient boosting. Ensemble methods are commonly used to reduce the risk over-fitting and increase the stability of output predictions. The training and testing performance with cross validation in the case with small training dataset is shown below. (*Note, please see the Response to Reviewer document for the figure.*) We can see that both random forest (red line) and gradient boosting (blue line) perform better (right graph) than simple linear regression (orange line) when the number of input locations are larger than 32 or 16, but they do not perform better than TTPCR (purple line) or PLS (green line), especially at the beginning of the test with few input locations. Moreover, it took much longer to train gradient boosting and random forest models (5 and 18 minutes) versus TTPCR (6 seconds). This makes these more complex models inappropriate for real-time training on visual field machines with limited computational power. Intuitively, linear models are well suited for visual field reconstruction since the visual field thresholds between different locations tend to be very strongly linearly correlated, so tree-based or other non-linear models are not the most appropriate choices.

Insert on line 372: “Similarly, we have also tried ensembled-based methods, such as random forest and gradient boosting, which also has the capability to reduce overfitting, but they did not offer better performance than TTPCR and PLS and took much longer to train.”

(*Note, please see the Response to Reviewer document for the figure.*)

Figure 1. (Left) Training and (Right) testing reconstruction error of various models. As the number of input locations approaches the total number of locations in the visual field (54), all models’ error approach zero.

 

Reviewer 2

>Reviewer #2: Perimetry, or visual field test, estimates differential light sensitivity thresholds across many locations in the visual field (e.g., 54 locations in the 24-2 grid). Recent developments have shown that an entire visual field may be relatively accurately reconstructed from measurements of a subset of these locations using a linear regression model. Here, authors have shown that incorporating a dimensionality reduction layer can improve the robustness of this reconstruction. Specifically, they proposed to use principal component analysis to transform the training dataset to a lower dimensional representation and then use this representation to reconstruct the visual field. they named the new reconstruction method the transformed-target principal component regression (TTPCR). When trained on a large dataset, the new method yielded results comparable with the original linear regression method, demonstrating that there is no underfitting associated with parameter reduction. However, when trained on a small dataset, our new method used on average 22% fewer trials to reach the same error. The results suggest that dimensionality reduction techniques can improve the robustness of visual field testing reconstruction algorithms. Authors must address my following queries

>

>• Presented values in figures and graphs must be statically verified.

We thank Reviewer #2 for the comments. 

We assume that the reviewer meant “statistically verified.” The main results shown in Fig 3 (formerly Fig 2) illustrating point-wise root-mean-square error vs number of trials are based on cross-validation results. Given the large number of simulations, variations in the results are small. For example, in Figure 3A (formerly 2A), the mean error±standard error for RMSE after 36 locations is 2.875±0.033 dB for LR, and 2.358±0.027 dB for the TTPCR, so this difference is already highly statistically significant (p<10-30). As such, we chose not plot the error bar as it would be too small. Finally, our method of presenting the results is line with what is typically done in the visual field simulation literature, including one paper published previously in PLOS ONE (Kucur et al 2017).

(*Note, please see the Response to Reviewer document for the figure.*)

Figure 2. Reproduction of Figure 3A (formerly 2A) with 95% confidence intervals in shaded color, which are extremely small and hard to notice on the graph. Therefore, for clarity, the original graph in the main text did not include this interval.

>• How the research used for comparison is selected in this manuscript, this is not clear

This paper improves upon the previously published, state-of-the-art paper about the Sequentially Optimized Reconstruction Strategy (SORS) by Kucur et al. [1] This is an important work in the field of visual field testing algorithms as it introduced a new paradigm of using machine learning and data-driven methods to automatically learn patterns from the visual field. Earlier methods incorporated certain rule-of-thumbs or heuristics. In addition, the SORS method is simple and easy to implement, and has been tested in a clinical population, such as by Hoehn et al. [2], whereas many of the other visual field testing algorithms are harder to implement and impractical in clinical settings. Therefore, the original SORS method is an important method for comparison. In this paper we extend this method to demonstrate that, in contrast to the original 2017 implementation which uses linear regression with a large number of weights, we can significantly reduce the number of parameters using dimensionality reduction techniques (e.g., PCA) while improving or maintaining the same performance in testing. This reduces the amount of training required and improves the efficiency and robustness of the algorithm. Please refer to lines 49-87 in the Introduction section for more details about the motivation of the paper.

[1] Kucur ŞS, Sznitman R. Sequentially optimized reconstruction strategy: a meta-strategy for perimetry testing. PLOS ONE. 2017 Oct 13;12(10):e0185049.

[2] Hoehn R, Häckel S, Kucur S, Iliev ME, Abegg M, Sznitman R. Evaluation of Sequentially Optimized Reconstruction Strategy in visual field testing in normal subjects and glaucoma patients. Investigative Ophthalmology & Visual Science. 2019 Jul 22;60(9):2477-.

Edit on line 83: “We incorporate these methods and compare the performance against existing methods (e.g., the original SORS linear regression method)”

>• How authors have selected the dataset in this research, it is not clear to readers

The Rotterdam dataset is the standard public glaucomatous visual field dataset in visual field testing literature. All visual field testing literature in the past decade uses this dataset which allows for comparison between different works. The dataset was chosen because it has a long history in literature and is high quality because it includes lots of repeated measurements of patients with varying degrees of glaucomatous defect severity. The original SORS paper used this dataset [1] and we follow the same convention here.

Insert on line 183: “… Rotterdam longitudinal glaucomatous visual field dataset, which is the same dataset used in previous work. [3, 15] The Rotterdam dataset encompasses a wide variety of glaucomatous patterns and has repeated measurements, ensuring the representativeness of the dataset to real-life conditions.”

>• Authors are suggested to cite and discuss

>‘ Zheng, M., Zhi, K., Zeng, J., Tian, C., & You, L. (2022). A hybrid CNN for image denoising. Journal of Artificial Intelligence and Technology, 2(3), 93-99.’

>‘ Zhang, Q., et al.: A robust deformed convolutional neural network (CNN) for image denoising. CAAI Trans. Intell. Technol. 1– 12 (2022). https://doi.org/10.1049/cit2.12110’

The task of visual field reconstruction involves taking a small number (e.g., 1, 2, 3, …) of threshold value inputs to predict a large number (54) of threshold value outputs. The figure below illustrates an example case of 16 input values. The few input values are sparsely sampled, and even in this example with 16 values, it is impossible to convolve with a 2x2 mask. In short, it is impossible to perform convolution on so many missing data. In addition, the visual field “image” is relatively small (a non-rectangular subset of a 9x8 grid) compared to conventional images which may benefit from using CNN to reduce the number of weights. The two papers suggested are related to denoising of relatively large images compared to the use case here, which is a fundamentally different from the task of reconstructing the kind of sparse, non-rectangular 9x8 visual field in this paper. 

We have clarified this by the following insertion.

Insert on line 374: “While convolutional neural network may seem like a natural choice for processing local visual field patterns, the sparseness of the SORS sampling process prohibits any convolution operations.”

(*Note, please see the Response to Reviewer document for the figure.*)

Figure 2. Illustration of the visual field reconstruction problem.

>• Comparison with the baseline deep learning models is required along with the comparison with baseline dimensionality reduction techniques

This paper improves upon the previously published paper about SORS. Therefore, the original SORS method, which uses linear regression, is used as a baseline for comparison. A simpler, robust model is desired because (1) visual field data tend to be noisy, (2) training data is limited, (3) interpretability is important for a medical algorithm, and (4) visual field machines tend to be offline, embedded devices with limited computational power. The TTPCR method proposed in this paper satisfies these goals. Deep learning models, such as deep auto-encoder, were discussed on lines 353-367 with performance illustrated in Figure S5. In short, we found the performance of deep learning methods to be worse than the linear models presented. Other researchers have also attempted to generalize the SORS model with deep learning approaches, requiring large amounts of training data without noticeably improving performance.[3] As a result, current implementations in the area still use the linear regression SORS model as a comparison baseline.[1,2]

[1] Kucur ŞS, Sznitman R. Sequentially optimized reconstruction strategy: a meta-strategy for perimetry testing. PLOS ONE. 2017 Oct 13;12(10):e0185049.

[2] Hoehn R, Häckel S, Kucur S, Iliev ME, Abegg M, Sznitman R. Evaluation of Sequentially Optimized Reconstruction Strategy in visual field testing in normal subjects and glaucoma patients. Investigative Ophthalmology & Visual Science. 2019 Jul 22;60(9):2477-.

[3] Kucur ŞS, Márquez-Neila P, Abegg M, Sznitman R. Patient-attentive sequential strategy for perimetry-based visual field acquisition. Medical image analysis. 2019 May 1;54:179-92.

Edit on line 362: “Although such deep, non-linear artificial-neural networks are generally more powerful, it did not provide superior performance for the visual field reconstruction task.”

Insert on line 372: “Similarly, we have also tried ensemble tree-based methods, such as random forest and gradient boosting, which may also overcome overfitting, but they did not offer better performance than TTPCR and PLS and took much longer to train.”

>• How are research variables selected for this manuscript?

Visual field testing, a psychophysics test, is a time-consuming process because it requires subjective response from the patient and samples many (54 or more) locations across the retina. While it is important to ensure accurate test results for accurate clinical diagnosis and monitoring, practically both patients and ophthalmologists also desire faster tests for patient comfort and clinic efficiency. Therefore, the research variables in this work are the number of trials to evaluate the test duration and point-wise root mean square error to evaluate the accuracy of the visual field estimate. A faster test (lower number of trials) to reach the same error would indicate a more efficient algorithm. Lower point-wise root-mean-square error (RMSE) is important as it is the basis for all down-stream clinical analysis. For example, mean deviation (MD) and visual field index (VFI) are two commonly used summary indices by ophthalmologists. These indices are derived from the raw visual field measurement data. Therefore, a lower point-wise RMSE would lead to more accurate visual field summary indices.

Insert on line 189: “Point-wise RMSE is used to evaluate as the accuracy of the result while number of trials is used for test duration.”

---

## [Decision Letter · Decision Letter 1]

17 Mar 2024

Improving the Robustness of The Sequentially Optimized Reconstruction Strategy (SORS) for Visual Field Testing

PONE-D-23-14760R1

Dear Dr. Shi,

We’re pleased to inform you that your manuscript has been judged scientifically suitable for publication and will be formally accepted for publication once it meets all outstanding technical requirements.

Kind regards,

Nouman Ali

Academic Editor

PLOS ONE

Additional Editor Comments (optional):

Reviewers' comments:

Reviewer's Responses to Questions

**Comments to the Author**

1. If the authors have adequately addressed your comments raised in a previous round of review and you feel that this manuscript is now acceptable for publication, you may indicate that here to bypass the “Comments to the Author” section, enter your conflict of interest statement in the “Confidential to Editor” section, and submit your "Accept" recommendation.

Reviewer #1: All comments have been addressed

Reviewer #2: All comments have been addressed

2. Is the manuscript technically sound, and do the data support the conclusions?

Reviewer #1: Yes

Reviewer #2: Yes

3. Has the statistical analysis been performed appropriately and rigorously? 

Reviewer #1: Yes

Reviewer #2: Yes

4. Have the authors made all data underlying the findings in their manuscript fully available?

Reviewer #1: Yes

Reviewer #2: Yes

5. Is the manuscript presented in an intelligible fashion and written in standard English?

Reviewer #1: Yes

Reviewer #2: Yes

6. Review Comments to the Author

Reviewer #1: In the revised manuscript, all my concerns were addressed. The models were clearly demonstrated in the current form and the extra experiment supported your conclusions well.

Reviewer #2: The quality of this submission is now improved and as a reviewer i am recommending the acceptance of this .

7. PLOS authors have the option to publish the peer review history of their article (what does this mean?). If published, this will include your full peer review and any attached files.

Reviewer #1: No

Reviewer #2: No

---

## [Editor Report · Acceptance letter]

26 Mar 2024

PONE-D-23-14760R1 

PLOS ONE

Dear Dr. Shi, 

I'm pleased to inform you that your manuscript has been deemed suitable for publication in PLOS ONE. Congratulations! Your manuscript is now being handed over to our production team.

Kind regards, 

on behalf of

Dr. Nouman Ali 

Academic Editor

PLOS ONE